# A Benchmark for Voice-Face Cross-Modal Matching and Retrieval

## Abstract

Cross-modal associations between a person's voice and face can be learned algorithmically, and this is a useful functionality in many audio and visual applications. The problem can be defined as two tasks: voice-face matching and retrieval. Recently, this topic has attracted much research attention, but it is still in its early stages of development, and evaluation protocols and test schemes need to be more standardized. Performance metrics for different subtasks are also scarce, and a benchmark for this problem needs to be established. In this paper, a baseline evaluation framework is proposed for voice-face matching and retrieval tasks. Test confidence is analyzed, and a confidence interval for estimated accuracy is proposed. Various state-of-the-art performances with high test confidence are achieved on a series of subtasks using the baseline method (called TriNet) included in this framework. The source code will be published along with the paper. The results of this study can provide a basis for future research on voice-face cross-modal learning.

## 1 Introduction

Studies in biology and neuroscience have shown that a person's appearance is associated with his or her voice (Smith et al., 2016b;a; Mavica & Barenholtz, 2013). Both the facial features and voice–controlling organs of individuals are affected by hormones and genetic information (Hollien & Moore, 1960; Thornhill & Møller, 1997; Kamachi et al., 2003; Wells et al., 2013), and human beings have the ability to recognize this association. For example, when speaking on the phone, we can guess the gender and approximate age of the person on the other end of the line. When watching a TV show without sound, we can also imagine the approximate voice of the protagonist by observing his or her face movements. With the recent advances in deep learning, face recognition models (Wen et al., 2016; Wu et al., 2018; Liu et al., 2017) and speaker recognition models (Wang et al., 2018; Li et al., 2017) have achieved extremely high precision. It is then natural to wonder if the associations between voices and faces could be discovered algorithmically by machines. The research on this problem could benefit many applications such as the synchronization of video faces with talking voices and the generation of faces according to voice.

In recent years, much research attention (Wen et al., 2018; Horiguchi et al., 2018; Nagrani et al., 2018a; Kim et al., 2018; Nagrani et al., 2018b) has been paid to voice-face cross-modal learning tasks, which has shown the feasibility of recognizing voice-face associations. This problem is generally formulated as a voice-face matching task and a voice-face retrieval task. The research on this problem is still at an early stage, and a benchmark for this problem still needs to be established. In this paper, we address this issue with the following contributions: 1) Existing methods are all evaluated on a single dataset of about 200 identities with limited tasks. The estimated accuracy always has great deviation due to the high sampling risk existed in cross-modal learning problem. Test confidence interval is proposed for qualifying the statistical significance of experimental results. 2) A solid baseline framework for voice-face matching and retrieval is also proposed. State-of-the-art performances on various voice-face matching and retrieval tasks are achieved on large-scale datasets with a high test confidence.

Table 1: Statistics of voice-face cross-modal datasets.Vox-VGG-n represents the combined dataset of the VoxCeleb (Nagrani et al., 2017; Chung et al., 2018) and VGGFace (Cao et al., 2018; Parkhi et al., 2015) and n denotes the version. The number of images refers to the number of all images remaining after MTCNN (Zhang et al., 2016) face detection.

| Dataset | #Identities | #Utterances | #Images |
| --- | --- | --- | --- |
| Vox-VGG-2 | 5,994 | 1,092,009 | 1,905,016 |
| Vox-VGG-1 | 1,251 | 153,516 | 573,283 |

## 2 RELATED WORKS

The existing methods for voice-face cross-modal learning can be classified as classification-based methods and pair-wise loss based methods, as shown in Figure 1. CNN-based networks are normally used to embed the voices and faces into feature vectors. SVHF (Nagrani et al., 2018b) is a prior study on voice-face cross-modal learning that investigated the performance of a CNN-based deep network on this problem. The human baseline for the voice-face matching task is also presented in this paper. DIMNet (Wen et al., 2018) learns a common representation for faces and voices by leveraging their relationships to some covariates such as gender and nationality. For pair-wise loss based methods, a pair or a triplet of vectors is embedded by a voice and face network, and contrastive loss (Hadsell et al., 2006) or triplet loss (Schroff et al., 2015) is used to supervise the learning of the embeddings. Horiguchi et al.'s method (Horiguchi et al., 2018) , Pins (Nagrani et al., 2018a), Kim et al.'s methods (Kim et al., 2018) are all these kind of methods. The aim of pair-wise loss based methods is to make the embeddings of positive pairs closer and the embeddings of negative pairs farther apart. In contrast, the aim of classification-based methods is to separate the embeddings of different classes. Of these two approaches, pair-wise loss based methods are better at distinguishing hard examples because of the characteristics of this approach.

There is still no related work which presents a benchmark for voice-face cross-modal learning tasks, which is addressed in detail as follows:

1) As for evaluation metrics, the reliability of experiments has not been addressed by all previous research. Test confidence is proposed in this paper. With the guidance of test confidence, reliable evaluations can be conducted.

2) As for tasks, joint matching and joint retrieval tasks established in this paper are not noticed by previous research. Though these tasks are direct extensions of traditional tasks, these very simple extensions can improve the performance of voice-face cross-modal learning dramatically.

3) As for models, the most similar work to TriNet of this paper is Kim et al.'s method (Kim et al., 2018). Both models use the triplet loss function. The main difference is that TriNet uses L2 normalization and voice-anchored embedding learning to constrain the feature space, because it is difficult to obtain satisfactory results by training directly in a huge Euclidean space. Though L2 normalization is a normal technique, it hasn't been introduced to the current problem.

4) As for datasets, currently available voice-face datasets are the data generated by the common speakers of VGGFace (Cao et al., 2018; Parkhi et al., 2015) face recognition dataset and VoxCeleb (Nagrani et al., 2017; Chung et al., 2018) speaker recognition dataset. As shown in Table 1, the voice-face datasets have two versions, Vox-VGG-1 and Vox-VGG-2, which include 1,251 and 5,994 identities, respectively. To the best of our knowledge, only Vox-VGG-1 is used in previous research. Both Vox-VGG-1 and Vox-VGG-2 are used to evaluate the proposed baseline method, TriNet.

## 3 TASKS AND EVALUATION

### 3.1 TASKS

**1:2 Matching and 1:n Matching.** Given an audio and two face candidates (only one of which is from the speaker of the audio), the goal is to find the face that belongs to the speaker. The more difficult l:n matching task is an extension of the 1:2 matching task that increases the number of candidate faces from 2 to $N$.

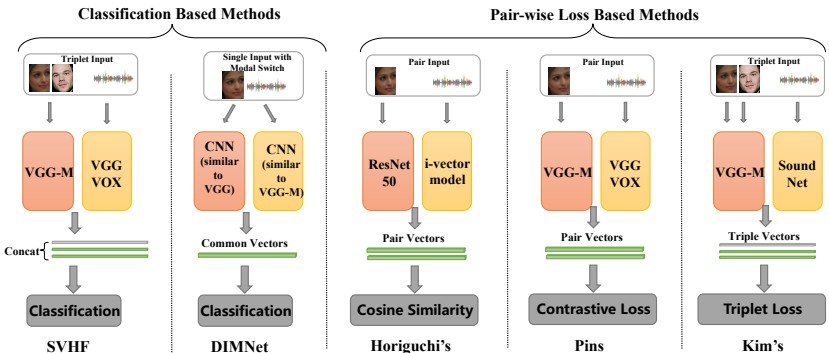

Figure 1: Comparison of existing methods.

**Retrieval.** Given a "query" voice, the goal of voice-face retrieval is to rank face images according to their relevance with respect to the voice query. This task is a supplement to the matching task, the position-related information of all retrieved faces is also effective for analyzing the model performance.

**Joint Matching and Joint Retrieval.** Instead of a single audio segment or single face for one identity, multiple audio segments and faces can provide more information. Matching and retrieval can be conducted on the mean embeddings of multiple audios or images. This is the simplest way to improve the performance of current voice-face matching and retrieval methods. Widespread video resources imply that the use of multiple faces and voices is feasible.

### 3.2 TEST CONFIDENCE

The evaluation criteria for matching and retrieval tasks are accuracy and mAP(Christopher et al., 2008) respectively. All previous studies (Wen et al., 2018; Nagrani et al., 2018a; Kim et al., 2018; Nagrani et al., 2018b) evaluated their methods on a single dataset of about 200 identities. As shown in the experiment (Section 5.4), the 1:2 matching accuracy tested on multiple datasets with 189 identities varies significantly, from 81% to 87%. ***So the results of all related works that used Vox-VGG-1 for training and testing are unreliable.*** Testing a model on a single small dataset may lead to a large deviation in the accuracy.

In 1:2 matching task, the accuracy is estimated on the sampled data, to represent the accuracy on the overall population. The estimated accuracy always has a large deviation due to the high sampling risk in the triplet sampling scenario.

Essentially, our aim is to obtain the correct matching probability of a single independent triplet. When the dataset and the model are determined, a single independent sampling conforms to the Bernoulli distribution $B(p)$. The results of $n$ samplings fit the binomial distribution $B(n, p)$. Interval estimation of a binomial distribution can be used for quantifying the deviation of the estimated accuracy. Suppose a dataset $D$ can generate up to $N$ triplets, and the number of sampled triplets used for testing is $n$. Among the sampled triplets, there are $m$ correctly matched triplets. Suppose the sample rate is $p$, where $p = \frac{m}{n}$, and the population rate of correctly matched $N$ triplets is $P$. When $n$ is sufficiently large, $p$ can be approximated as normal distribution $p \sim N(P, \frac{P(1-P)}{n})$. By converting it to a standard normal distribution, we obtain $u = \frac{p-P}{\sqrt{\frac{P(1-P)}{n}}} \sim N(0, 1)$. For a significance level $\alpha$, the confidence interval of $p$ is $(p - u_{\frac{\alpha}{2}}\sqrt{\frac{p(1-p)}{n}}, p + u_{\frac{\alpha}{2}}\sqrt{\frac{p(1-p)}{n}})$. Testing a model on multiple datasets is strongly recommended when the dataset is very small. The test can be performed multiple times on datasets with a similar scale, and the results are regarded as conforming to the normal distribution. The t-test can then be used to estimate the confidence interval of the accuracy.

## 4    TriNet Baseline Method

As shown in Figure 2, the baseline method in the proposed framework consists of three steps: extracting voice and face features, constraining embeddings to a spherical space, and computing the triplet loss. After training, the face embedding and voice embedding form their own regions, and the distance between positive samples tends to be smaller.

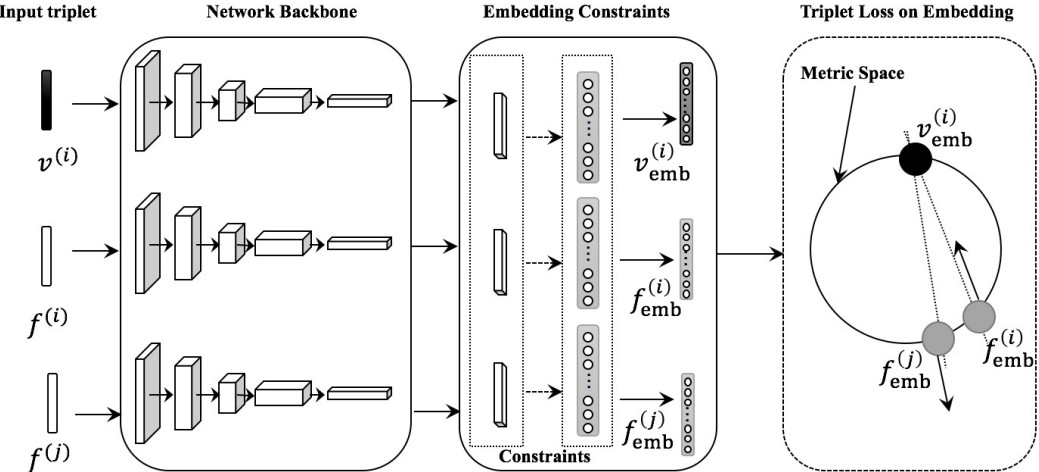

Figure 2: Overview of TriNet. 1) A feature extraction module extracts voice and face features. 2) Embeddings are constrained to lie on a spherical space using L2 normalization. In this step, the audio network is frozen. 3) The triplet loss function is used to learn the embeddings.

### 4.1    Triplet Mining

The input triplets for the embedding network need to be mined from the datasets, the number of which is extremely large. In previous research (Nagrani et al., 2018b; Kim et al., 2018; Wen et al., 2018; Horiguchi et al., 2018), discrete triplets are randomly mined to create a single input each time, which will lead to training and test inefficiency. Identity based sampling named as online mining is adopted in this paper, which can greatly improve the training and testing efficiency. In the identity based sampling, a batch of identities is randomly selected first, and then certain number of face images and audios for each identity of the batch are sampled. Triplets are generated based on each batch of identities. Triplet Loss is susceptible to noise which means the direction of network convergence is easy to be changed by few noise samples. Identity based training can effectively handle the disadvantage of Triplet Loss.

### 4.2    Embedding Constraints and the Loss Function

For a specific triplet $< v^{(i)}, f^{(i)}, f^{(j)} >$, $v^{(i)}$, and $f^{(i)}$ are from the same identity, and $v^{(i)}$ and $f^{(j)}$ are from different identities. The feature extraction functions for voice and face are defined as $Feature_v(v)$ and $Feature_f(f)$, respectively, and a fully connected layer is added to form the embedding vectors as $emb_v(v) = s \times \|W_v \times Feature_v(v) + B_v\|_2^2$ and $emb_f(f) = s \times \|W_f \times Feature_f(f) + B_f\|_2^2$. LResNet50 (He et al., 2015) and Thin ResNet34 (Xie et al., 2019) with NetVLAD (Arandjelovic et al., 2017) are networks that perform well on face recognition and speaker recognition tasks, respectively. These two networks are used in this paper for face feature extraction and voice feature extraction.

As illustrated in Figure 3a, embedded vectors from the same person will appear in a Euclidean space after a long period of training. Because there are billions of input triples, it is difficult to obtain satisfactory results by directly training in a huge Euclidean space. To deal with this problem, two strategies are adopted in the method proposed in this paper. First, L2 normalization is added to constrain the embedding vectors to a spherical space (Figure 3b). Second, voice-anchored embedding learning is adopted. By freezing the pre-trained voice embedding network, feature vectors

from voice serve as anchors, and the goal of the model is to make positive instances approach each other while keeping the negative instances away (Figure 3c). Examples tend to be distinguished much better and faster in the voice-anchored embedding learning process when used with the L2 constrained space. Triplet loss is adopted in this paper for embedding learning. Suppose $d(x, y)$ indicates Euclidean distance; the loss function is defined as

$$Loss = \sum_{v^{(i)}, f^{(i)}, f^{(j)}, i \neq j} \max \{ d(emb_v(v^{(i)}), emb_f(f^{(i)})) - d(emb_v(v^{(i)}), emb_f(f^{(j)})) + m, 0 \}, \tag{1}$$

where $m$ is a margin to control the distance between positive and negative pairs.

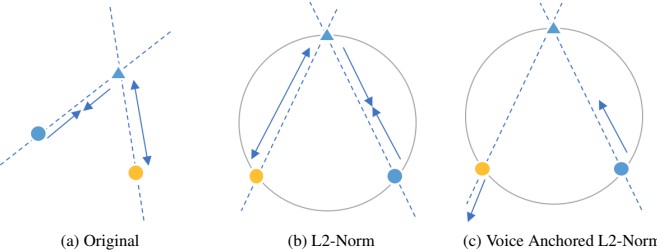

(a) Original        (b) L2-Norm        (c) Voice Anchored L2-Norm

Figure 3: Visualization of three metric learning methods.

## 5 EXPERIMENT

### 5.1 BASELINE MODEL SETUP

**Training Details.** TriNet was trained on Vox-VGG-2 with 5,994 identities. Face detection based on MTCNN (Zhang et al., 2016) was conducted and all face images were then rescaled to $112 \times 112 \times 3$ to form the input for the face embedding networks. Audio preprocessing consisted of 512-point fast Fourier transform, a short-time Fourier transform (Benesty et al., 2011) for each frame, and normalization. The audio segments used for training were uniformly trimmed to 2.5 s for training efficiency, and the test audio segments were not clipped. The input shape for a $k$-s audio clip is $257 \times (100 \times k) \times 1$. The voice embedding network and face embedding network were pre-trained by VoxCeleb2 and VGGFace2, respectively. Margin, $m$, for the triplet loss was set to 1, and scale, $s$, for the L2 normalization was set to 128. The Adam optimizer was adopted in these experiments, and the total number of learning steps was 70k. The learning rates of the final fully connected layer for $step < 20k$, $20k < step < 40k$, $40k < step < 60k$, and $step > 60k$ were $10^{-3}, 10^{-4}, 10^{-5}$, and $10^{-6}$ respectively. The learning rate of the face embedding network was fixed to $10^{-6}$.

**Testing Details.** 1) For the 1:2 matching task, a total of 10,000 steps were tested on the baseline, which implies that a total of 30.72 million triples were tested. Note that the gender of the test triples in the 1:2 matching task was balanced. 2) For the 1:n matching task, the number of tuples to be sampled will be much higher than the number of triples in 1:2 matching; therefore, we performed this test directly on the 10k tuples. This will lower the confidence level, but the results are still useful for comparisons. 3) For the retrieval task, a face database of 500 pictures was constructed from 100 randomly selected identities, and 40 audio queries were constructed for each identity.

Table 2: Comparison of the proposed method with other models on the 1:2 matching task and retrieval task. "ACC" is accuracy.

| Tasks | Method | Test Identities | Test Triplets | ACC(%) |
|---|---|---|---|---|
| 1:2 Matching | TriNet | 1,251 | 30M | $84.48 \pm 0.01\%$ |
| | SVHF(Nagrani et al., 2018b) | 189 | 10k | $81.00 \pm 1.01\%$ |
| | DIMNet-IG(Wen et al., 2018) | 189 | 678M | 84.12% |
| | Kim's (Kim et al., 2018) | 250 | - | 78.20% |
| | Horiguchi's (Horiguchi et al., 2018) | 216 | 38B | 77.80% |
| | **Method** | **Test Identities** | **Chance (mAP%)** | **mAP(%)** |
| Retrieval | TriNet | 1251 | 2.15 | 11.48 |
| | DIMNet-IG (Wen et al., 2018) | 189 | 1.07 | 4.42 |
| | Horiguchi's (Horiguchi et al., 2018) | 216 | 0.46 | 1.96 |

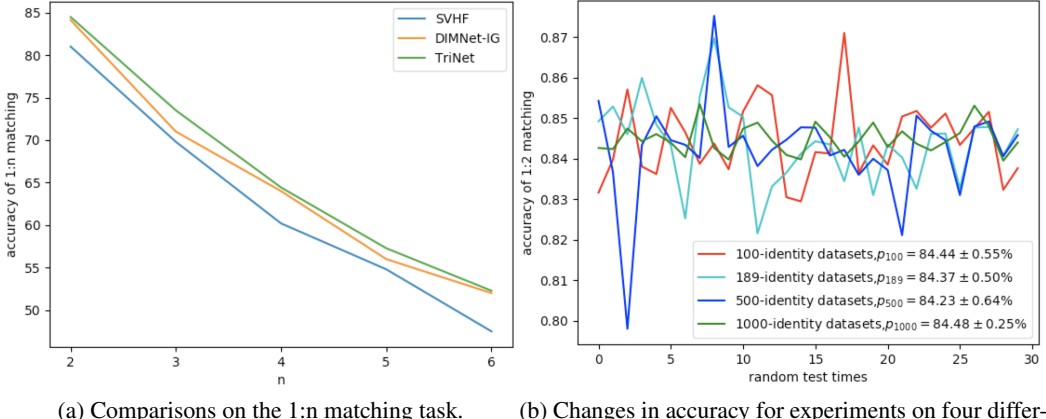

(a) Comparisons on the 1:n matching task.

(b) Changes in accuracy for experiments on four different sizes of datasets.

Figure 4: (a) Comparisons on the 1:n matching task. (b) Accuracy curves.

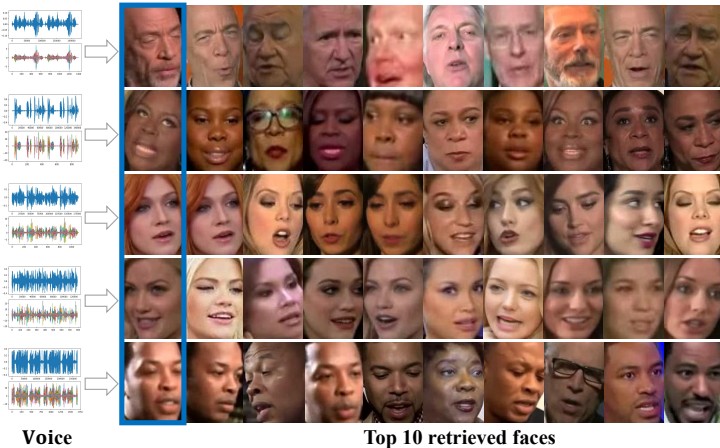

**Voice**          **Top 10 retrieved faces**

Figure 5: Qualitative analysis of retrieval results produced by TriNet.

## 5.2 COMPARISONS ON MATCHING AND RETRIEVAL TASKS

Comparisons of TriNet and related works on the 1:2 matching task and retrieval task are shown in Table 2. TriNet achieves state-of-the-art performance on these two main tasks. As shown in Figure 4a, on the 1:n task, the performances of all the methods decrease rapidly as n increases. This task is still significantly difficult. Some TriNet retrieval results that fit $p@1 = 1$ are illustrated in Figure 5. The top ranked faces in each sequence are very similar to the target face.

## 5.3 JOINT TASKS PERFORMANCE

The results of 1:2 joint matching using mean voice and mean face are shown in Table 3. Two variables, $m_f$ and $m_v$, are introduced to represent the number of faces and audios used to compute the mean embedding. Various values of $m_f$ and $m_v$ are tested for 1:2 matching. For retrieval, $m_v$ was set to 20 and $m_f$ was set to 5. This simple strategy of using multiple faces and voices can further improve the matching accuracy and retrieval mAP. Specifically, when $m_v = m_f = 10$, an accuracy of $89.66 \pm 0.80\%$ can be obtained for TriNet on the 1:2 matching task, which is 5% higher than that of single voice and single face matching. This improvement reveals a broad prospect for future research on using video data for cross-modal learning.

Table 3: Performance of 1:2 joint matching and joint retrieval using mean face and mean voice embeddings. Here, $m_f$ and $m_v$ denote the number of face images and audio segments, respectively, used to calculate the averages. The values of $m_v$ and $m_f$ are 1 by default.

| Joint Matching Using Mean Voices and Mean Faces (TriNet) | | | | | |
|---|---|---|---|---|---|
| $m_f(m_v = 1)$ | ACC(%) | $m_v(m_f = 1)$ | ACC(%) | $m_f + m_v$ | ACC(%) |
| 5 | 85.55 | 5 | 85.13 | 1+1 | 84.48 |
| 10 | 84.56 | 10 | 86.01 | 5+5 | 86.42 |
| 15 | 86.28 | 15 | 85.49 | 10+10 | **89.66** |
| 20 | 86.53 | 20 | 86.16 | 20+20 | 89.54 |
| 30 | 84.71 | 30 | 85.63 | 30+30 | 89.28 |
| Joint Retrieval Using Mean Voice and Mean Face | | | | | |
| | | mean voice mAP(%) | | mean face mAP(%) | |
| TriNet | | 12.53 | | 21.65 | |
| Chance | | 2.15 | | 5.25 | |

Table 4: Effect of training set scale. "TriNet" used the default configuration. "TriNet-train1000" was trained on 1,000 and tested on 189 identities.

| Method | #Train Identities | #Test Identities | #Test Triplets | ACC(%) | mAP(%) |
|---|---|---|---|---|---|
| TriNet | 5994 | 1251 | 30M | $84.48 \pm 0.01\%$ | 11.48% |
| TriNet-train1000 | 1000 | 189 | 3M | $83.55 \pm 0.55\%$ | 9.61% |

## 5.4 Test Confidence of Datasets with Different Scales

Figure 4b shows the fluctuations in the estimated accuracies of TriNet on the 1:2 matching task when 30 repeated random tests were conducted. The numbers of sampled identities for each curve are 100, 189, 500, and 1,000. For a determined dataset scale (such as 100 identities), instead of testing the model on a single dataset with 100 identities, 30 randomly sampled sets of data with 100 identities were used for testing. As shown in the figure, when a small-scale dataset is used, the accuracy of different runs fluctuates substantially. For large datasets, fluctuations in test accuracies also exist; however, in general, the test results are more generalized; therefore, large datasets are strongly recommended for evaluation.

## 5.5 Ablation Study

There are various options in the baseline model. To determine the option that has a greater impact on performance, we conducted a more detailed ablation study.

### 5.5.1 Training Scale

The size of the training dataset identities used by the baseline is five times that of most related studies. To demonstrate the effects of a larger training dataset on the results, TriNet was also trained on a dataset of 1,000 identities and tested on a dataset with 189 identities. As shown in Table 4, the improvement of training on large scale dataset is near 1%. The upper limit of the results is similar to those of DIMNet-IG. Adding more identities can increase the performance by 0.5%. It is difficult to further improve the performance by increasing the size of the dataset. In contrast, integrating multiple faces and voices is an effective way to further improve the performance.

### 5.5.2 Preprocessing

We need to study whether face detection should be used and how big the detection box should be. As the results in Table 5 reveal, without the use of face detection, a large amount of noise is introduced along with a few useful features, and the performance of the baseline model on all matching and retrieval tasks decreases. When the size of the default detection box is increased by 1.1 times, better performance is obtained.

### 5.5.3 Network Structure

As shown in Table 5, deeper CNN structures such as SE-ResNet50 (Hu et al., 2018) and the structure used in DIMNet outperform traditional shallow structures such as VGG-M. An SE-ResNet50 with a

Table 5: Ablation study of TriNet. 1:2(%) and 1:3(%) denote accuracy for 1:2 and 1:3 matching tasks, and Male(%) and Female(%)denote accuracy on all male triplets and female triplets, respectively.

| Settings | | 1:2(%) | 1:3(%) | Male(%) | Female(%) | mAP(%) |
|---|---|---|---|---|---|---|
| TriNet (Default) | | 84.48 | 73.50 | 69.85 | 71.04 | 11.48 |
| **Detect Scale** | **Audio Crop** | | | | | |
| Without Scale | Whole | 83.04 | 71.13 | 66.99 | 69.69 | 8.22 |
| 1.1 Scale | Whole | 84.27 | 73.28 | 70.03 | 71.65 | 10.42 |
| 1.0 Scale | Crop 2.5 s | 83.11 | 70.49 | 61.41 | 69.45 | 9.09 |
| **Face Backbone** | | | | | | |
| VGG-M | | 81.30 | 68.95 | 64.90 | 64.84 | 8.06 |
| Face Backbone of DIMNet | | 83.88 | 72.06 | 62.93 | 68.98 | 10.05 |
| SE-ResNet50 | | 84.0 | 72.62 | 69.22 | 69.49 | 9.40 |
| **Face Frozen** | **Voice Frozen** | | | | | |
| N | N | 84.43 | 73.56 | 68.27 | 68.85 | 9.77 |
| Y | N | 81.68 | 68.52 | 64.87 | 66.76 | 8.33 |
| **Metric Space** | **Metric Scale** | | | | | |
| Euclidean | 128 | 82.19 | 69.29 | 69.38 | 64.97 | 8.44 |
| L2 | 1 | 81.71 | 69.09 | 62.62 | 67.49 | 7.58 |
| L2 | 512 | 84.27 | 72.48 | 69.43 | 71.77 | 10.02 |
| **Pre-Face** | **Pre-Voice** | | | | | |
| MS-1B(Guo et al., 2016) | VOX2 | 84.13 | 71.71 | 63.81 | 74.70 | 10.47 |
| None | VOX2 | 73.29 | 58.20 | 54.26 | 58.90 | 5.56 |
| None | None | 70.64 | 54.11 | 52.21 | 51.47 | 3.91 |

squeeze-and-excitation module does not produce better results than the original ResNet50 structure used in the baseline model.

### 5.5.4 EMBEDDING CONSTRAINTS

The effects of using L2 normalization and freezing the pre-trained networks are analyzed here. As presented in Table 5, using L2 normalization improves the performance of 1:2 matching accuracy by 2% and mAP by 2%. In the default configuration, the size of the metric space is 128. The model performance decreases when the scale is set to 1, which indicates that it is necessary to properly increase the size of the metric space. Freezing the face embedding network reduces the performance, whereas freezing the voice embedding network improves the performance slightly. This is because human voices are only related to some local features of human faces, and similar faces in traditional face recognition tasks do not necessarily have similar voices. Therefore, voice-anchored embedding learning outperforms face-anchored embedding learning. Training efficiency is improved substantially by freezing the voice network.

### 5.5.5 PRE-TRAINING

As shown in Table 5, when TriNet is pre-trained on the large dataset MS-1B (Guo et al., 2016), its performance is not improved. However, without pre-training, the model's performance is substantially reduced. (Note that in this case, the voice network was not frozen.)

## 6 CONCLUSION

In this study, a benchmark was established for voice-face matching and retrieval. The contributions of this paper are as follows. A solid voice-face matching and retrieval baseline method (TriNet) was proposed, which was tested on large-scale dataset with comprehensive ablation studies. The test confidence was proposed as a metric for qualifying the statistical significance of the experiments. On the 1:2 matching and retrieval tasks, TriNet achieved an accuracy of 84.48% and a mAP of 11%. Compared with the best results published so far, there is a 7% improvement in mAP. Using mean face and mean voice embeddings, the matching accuracy and retrieval mAP can be further improved by approximately 5% and 10%, respectively. This improvement implies a broad prospect for future research on using video data for cross-modal learning.

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
