# OpenReview forum: "A Benchmark for Voice-Face Cross-Modal Matching and  Retrieval"
_ICLR.cc/2021/Conference — Reject_

### Official Review · AnonReviewer3 · 2020-10-26
**The main contributions (training on more data) and L2 normalising the embeddings of a network are not sufficient for acceptance to ICLR.**

**Rating:** 3
**Confidence:** 5

**Review:**

The goal of this paper is to learn cross-modal associations between a person’s face and a voice. The authors use a standard three stream network trained with a triplet loss, and evaluate on the VoxCeleb-VGGFace datasets.

Strengths:
- The authors train on VoxCeleb-VGGFace2, and evaluate on VoxCeleb1, which is a larger set of identities, and perform a t-test to show statistical significance.

- The ablation study showing that performance saturates with more training data is interesting.


Weaknesses:
- There is limited novelty in the method. The authors use the triplet loss which has been widely used for this problem before (Kim et al. 2018, Cheng et al. 20- https://dl.acm.org/doi/pdf/10.1145/3394171.3413710). The authors also claim that anchoring the voice subnetwork with frozen weights is a novel contribution but don’t discuss that this teacher-student style model was already tried in the Learnable PINs paper (where the face subnetwork is frozen). L2 normalisation of embeddings has also been used widely with the triplet loss, and has also been used in Learnable PINs (and hence been applied to this problem as well).

- Table 2 is not a fair comparison. The authors have compared the performance of different models on completely different test sets, with a different number of speakers! To assess the performance of TriNet, it must be compared to SOTA methods on the same test set of 189 speakers, or if the authors prefer - on all of the 1,251 speakers in VoxCeleb1. In this case the other SOTA methods must be re-evaluated on this new test set.

---

### Official Review · AnonReviewer4 · 2020-10-27
**Review for A BENCHMARK FOR VOICE-FACE CROSS-MODAL MATCHING AND RETRIEVAL**

**Rating:** 3
**Confidence:** 5

**Review:**

This work focuses on the problem of cross-modal matching and retrieval for face and voice modalities. The paper suggests a new benchmark for the evaluation of both matching and retrieval tasks. It also proposes a confidence margin computation to verify the statistical significance of the results. The results reported on Vox (voice) and VGG (face) dataset using the suggested benchmark are encouraging. I have summarized my comments below which will help in improving the quality of this manuscript:

1. I believe the technical contribution of this paper is very limited for ICLR. This work builds upon the previous works in the area of cross-modal retrieval. The contribution is very incremental and most of the conclusions are very well known. The multi-stream N/W architecture is very similar to the previous works and triplet loss has been used extensively for face and voice tasks in the past. The suggested protocol just uses a different combination of training and evaluation set from Vox and VGG sets.

2. In section 3.2, the authors mentioned "So the results of all related works that used VoxVGG-1 for training and testing are unreliable." I feel this is a very strong statement and may not always be true. First, the Vox dataset is collected from YouTube videos and represents a wide variety of conditions and results on the careful partition (no overlapping identities and conditions) of training and test may be very helpful. Second, in cases where evaluation conditions or use cases are known and similar to the VoxVGG-1 this may not hold true.

3. In speaker recognition, data augmentation plays a huge role in learning a noise-robust representation. It is not clear if the authors applied data augmentation to the Vox2 training set. These works highlight the importance of data augmentation:
Snyder, D., Garcia-Romero, D., Sell, G., Povey, D. and Khudanpur, S., 2018, April. X-vectors: Robust DNN embeddings for speaker recognition. In 2018 IEEE International Conference on Acoustics, Speech and Signal Processing (ICASSP) (pp. 5329-5333). IEEE.
Zeinali, H., Wang, S., Silnova, A., Matějka, P. and Plchot, O., 2019. But system description to voxceleb speaker recognition challenge 2019. arXiv preprint arXiv:1910.12592.

4. Did the authors use any voice activity detection system to remove silence in the audio?

5. In section 5.4, the authors claim "Therefore, voice anchored embedding learning outperforms face-anchored embedding learning." It would be better to clarify these results in Table 5.

6. The current state-of-the-art in speaker recognition is TDNN based x-vector. I would request authors to add a reference to the following paper in the introduction:
Snyder, D., Garcia-Romero, D., Sell, G., Povey, D. and Khudanpur, S., 2018, April. X-vectors: Robust DNN embeddings for speaker recognition. In 2018 IEEE International Conference on Acoustics, Speech and Signal Processing (ICASSP) (pp. 5329-5333). IEEE.

---

### Official Review · AnonReviewer2 · 2020-10-28
**technical contribution is limited**

**Rating:** 4
**Confidence:** 4

**Review:**

Summary:
This paper aims to propose a benchmark for voce-face matching and retrieval problem. As shown by the test confidence analysis, the model is suggested to be evaluated on a large dataset or multiple datasets to avoid the large deviation in the accuracy. A baseline method TriNet and joint matching & retrieval are proposed. Improved results are reported in the experiment section.

My biggest concern is the unclear contribution of this paper.
Test confidence is proposed, but has not been used to measure any results reported in this paper. It seems a bit disconnected. The only conclusion from test confidence analysis is to test model on multiple dataset, which is obvious.

The TriNet uses L2 normalized triplet loss, which is also not new and can be found in many previous work, e.g. [1]. Simply applying this normalized triplet loss to cross-modal matching is not a significant contribution.
[1] Schroff, Florian, Dmitry Kalenichenko, and James Philbin. "Facenet: A unified embedding for face recognition and clustering." Proceedings of the IEEE conference on computer vision and pattern recognition. 2015.

Also, the results in Table 2 are not fair comparison. As the paper points out, the training and testing data of TriNet are different from other methods, which means the results are not comparable. So I couldn't see any insight from the results.

---

### Decision · Program_Chairs · 2021-01-07
**Final Decision**

**Decision:**

Reject

**Comment:**

The reviewers pointed out several opportunities for improvements and concurred that the paper needs significant work before it is ready for publication.  The authors did not provide a rebuttal. We hope the review process was useful to the authors.